# *Elemental*: An Open-Source Wireless Hardware and Software Platform for Building Energy and Indoor Environmental Monitoring and Control

**DOI:** 10.3390/s19184017

**Published:** 2019-09-18

**Authors:** Akram Syed Ali, Christopher Coté, Mohammad Heidarinejad, Brent Stephens

**Affiliations:** 1Department of Civil, Architectural, and Environmental Engineering, Illinois Institute of Technology, Chicago, IL 60616, USA; muh182@iit.edu; 2Entropealabs, Chicago, IL 60626, USA; chris@entropealabs.com

**Keywords:** open-source, IAQ monitoring, indoor environment, building energy consumption, hardware, software

## Abstract

This work demonstrates an open-source hardware and software platform for monitoring the performance of buildings, called *Elemental*, that is designed to provide data on indoor environmental quality, energy usage, HVAC operation, and other factors to its users. It combines: (i) custom printed circuit boards (PCBs) with RFM69 frequency shift keying (FSK) radio frequency (RF) transceivers for wireless sensors, control nodes, and USB gateway, (ii) a Raspberry Pi 3B with custom firmware acting as either a centralized or distributed backhaul, and (iii) a custom dockerized application for the backend called *Brood* that serves as the director software managing message brokering via Message Queuing Telemetry Transport (MQTT) protocol using VerneMQ, database storage using InfluxDB, and data visualization using Grafana. The platform is built around the idea of a private, secure, and open technology for the built environment. Among its many applications, the platform allows occupants to investigate anomalies in energy usage, environmental quality, and thermal performance via a comprehensive dashboard with rich querying capabilities. It also includes multiple frontends to view and analyze building activity data, which can be used directly in building controls or to provide recommendations on how to increase operational efficiency or improve operating conditions. Here, we demonstrate three distinct applications of the *Elemental* platform, including: (1) deployment in a research lab for long-term data collection and automated analysis, (2) use as a full-home energy and environmental monitoring solution, and (3) fault and anomaly detection and diagnostics of individual building systems at the zone-level. Through these applications we demonstrate that the platform allows easy and virtually unlimited datalogging, monitoring, and analysis of real-time sensor data with low setup costs. Low-power sensor nodes placed in abundance in a building can also provide precise and immediate fault-detection, allowing for tuning equipment for more efficient operation and faster maintenance during the lifetime of the building.

## 1. Introduction

Understanding the complex relationships between occupant activity, air quality, energy usage, and occupant comfort levels in buildings requires monitoring many subsystems in addition to the perceived comfort of occupants [1,2,3]. Traditionally, these parameters are monitored by hardware and software that are expensive, proprietary, and often limited in terms of ease of use and flexibility [4]. Adding data monitoring and visualization layers to buildings can provide a glimpse into its energy use, thermal performance, daily operation, and maintenance requirements. Visualizing basic environmental and energy use data is already available in many existing building energy management systems. However, these systems typically do not utilize real-time activity data within buildings to inform their control strategies, nor do they make that data easily available for analysis, particularly in smaller residential and commercial buildings for which building energy management systems are often prohibitively expensive [5]. This results in unsatisfactory thermal comfort for occupants, inefficient usage of energy (especially during unoccupied periods), and a lack of zone-level anomaly detection for buildings and systems.

In the past decade, an exponential rise in open-source hardware and software technologies have facilitated the development and implementation of low-power monitoring systems for numerous applications, enabling data monitoring and controls in a cost-effective and hyper-local manner. With the advent of Internet of Things (IoT) and commercialization of IoT products, inexpensive wireless sensors have also seen a large adaptation in residential buildings around the world. Wireless sensor networks for indoor environmental monitoring have extensively proved the need for such localized data monitoring [6,7,8,9,10,11,12,13,14,15,16,17,18,19]. However, prohibitive upfront costs and a lack of standards related to implementation of IoT technologies in buildings has led to isolated adoption of IoT technologies that often cannot be easily integrated with other systems or replicated when needed [20]. Many existing buildings remain unmonitored or poorly monitored, leaving many opportunities for energy savings and improving indoor environmental conditions unaddressed.

This paper proposes an open-source hardware and software platform, called the *Elemental* platform, that enables real-time and historical analysis of a building’s performance and thermal comfort of occupants within by: (1) integrating accurate, inexpensive, low power wireless sensors and controls; (2) connecting existing commercial IoT devices, heating, ventilation, and air-conditioning (HVAC) controls, and energy monitoring systems; (3) and providing simple endpoints to allow building systems and control devices to access granular zone-level environmental and building activity information to make smarter decisions. The goal of the platform is to standardize all building information and controls in one place, while being fully open-source to allow large scale implementation. This project builds on an open-source sensor platform previously developed by our research group (OSBSS: Open Source Building Science Sensors [18]) and recently expanded upon in collaboration with the National Association of Realtors [21].

## 2. Materials and Methods

Continuous advancements in materials science and micro-scale manufacturing techniques has led to the development of highly power-efficient radios for wireless communication and low-power microprocessors that enable long-term embedded applications. Additionally, major developments in networking, computing power, and digital storage densities allow virtually unlimited data storage and high-speed network communication, enabling near-instant data retrieval and analysis. However, these advancements are not yet fully realized in existing indoor data collection and smart home/building applications.

### 2.1. Existing Technologies

Data collection for scientific research and smart home projects is increasingly being accomplished using low power microcontrollers and various open-source platforms, attributable in large part to their widespread availability, wide community support, and ease of use. For example, wireless sensor platforms based on Arduino and/or Raspberry Pi boards have successfully been used to monitor indoor and outdoor environmental quality [6,7,22,23,24,25,26] and building energy use [27]. However, projects based on off-the-shelf sensors and hardware such as Arduino boards commonly have their limitations. For example, they have relatively high-power consumption, which requires external power to be available [28]. This limits battery usage, thereby limiting the applications for which they can be most useful. The setups can also be unreliable due to the use of jumper wires to connect components, which can be fragile and loose. They have limited data storage without expensive add-ons. Additionally, inexpensive sensors that are commonly used with Raspberry Pi and Arduino-based boards usually have inadequate accuracy for research or control-based applications. The cost of hardware increases sharply when scaling up the setup for deployment on a larger scale. Using various types of open-source and commercial sensors and devices for a project also has the problem of data fragmentation, in which there is no central location for data storage and each platform produces data in their own format. This requires additional steps to standardize the data for analysis which takes a lot of time. There are many other challenges around the use of low-cost sensors such as custom software development and setup, which can sometimes have a steep learning curve and restrict engineers who are seeking a quick way to deploy a wireless sensor network for monitoring and control.

### 2.2. Custom PCBs

To address the aforementioned issues with existing technologies, we developed custom printed circuit boards (PCBs) for measuring common indoor environmental and building operational parameters, including air and surface temperature, relative humidity, light intensity, barometric pressure, motion activity, carbon dioxide (CO_2_) concentrations, particulate matter, total volatile organic compounds (TVOCs), and analog voltage inputs from other devices such as current transducers, heat flux sensors, and a wide variety of environmental sensors. We also developed custom PCBs that can enable knob and actuator controls for building systems such as radiator controllers, variable air volume (VAV) boxes, and various valves. The idea behind developing these boards is to provide a quick way to setup wireless sensors and controls in buildings which use a standard data format that can expedite analysis and control in building applications.

Figure 1a shows a basic block diagram of a sensor and control node. This includes a low-power microprocessor, an RFM69 radio module for wireless communication, a sensor or control device, an optional switch which can deactivate the sensors when needed for achieving lower power consumption, and a power source, which is typically a lithium ion polymer battery or 5 V power either from USB or any commercial power adapter.

Figure 1b illustrates the network architecture of a typical wireless sensor network. All sensor nodes and control nodes have their own unique node ID for identification. Up to 1023 nodes can be used in a network of sensors deployed in a building, however, it is also possible to use different networks for different applications, thereby increasing the total number of nodes that can physically coexist inside a building. Limitations in actual deployments will depend on communication intervals, transmission speeds, network congestion, size of data packets, physical obstructions and building materials, indoor environmental conditions, and other factors. However, at least several hundreds of nodes can be easily deployed in a large building with careful planning. All nodes communicate with a gateway that relays information directly to a host computer, in this case, a Raspberry Pi. The Raspberry Pi manages the data, either by storing it locally, or sending it to a local server or to a cloud-based backend for detailed data analysis and real-time data monitoring. The gateway, backhaul, and backend portion of the architecture is described in more detail in Section 3.

Figure 2a–c show several low-power wireless sensor nodes that we have developed and built. These can measure temperature, relative humidity, light intensity, motion, and door/window opening. These boards have an average idle current draw of around 6.5 µA and can last up to 2 years on a single 2000 mAh lithium-ion polymer battery logging at 1-min intervals and not subjected to wide swings in temperature (the actual battery life of a deployed sensor node with a 1200 mAh battery is shown later in Section 4.1). Figure 2d shows a wireless sensor board that measures CO_2_ concentration. This board requires external 5 V power due to the inherent nature of the CO_2_ measurement type used in the sensor (non-diffusive infrared). Keeping the sensor active at all times allows the sensor to maintain high accuracy, coupled with its built-in automatic background calibration feature helps correct for long-term drifts. Figure 2e shows an custom indoor air quality (IAQ) monitor that can measure temperature, relative humidity, light intensity, CO_2_ and TVOC concentrations, surface temperature, barometric pressure, particulate matter (i.e., approximations of PM_2.5_ and PM_10_ via a low-cost optical sensor), and external 0–5 V output from any commercial sensor, device, or actuator. This device is carefully designed for accurate measurements and has a flash memory on-board that enables data storage and over-the-air firmware updates when needed. Wireless firmware updates are accomplished using LowPowerLab’s wireless OTA software [29]. The accuracy of the sensors used on these boards were previously verified against similar research- and commercial-grade sensors commonly used in building applications [18,30], similar to procedures that have been used in other studies [31]. The sensors themselves, which are listed in Table 1, are commonly used in the industry for high accuracy environmental monitoring. Figure 2f shows a USB gateway for the sensor nodes. This device can be plugged into a USB-A port of any computer. It receives wireless data from all sensor node types and sends it over serial to the host computer at a baud rate of 115200. This device can also act as a transmitter for sending commands to control nodes based on the RFM69 radio. In all examples in Figure 2, the custom PCBs are aimed to make large scale deployments easy and cost-effective.

All boards use RFM69 wireless radios by HopeRF (Shenzhen, China). The RFM69 is a highly integrated Frequency Shift Keying (FSK) RF transceiver capable of operation over a wide frequency range, including the 433, 868 and 915 MHz license-free Industry, Scientific and Medical (ISM) frequency bands [32]. This allows the sensor nodes to operate independently in their own sub-GHz wireless networks, removing dependence on common wireless protocols that operate in the exceedingly crowded 2.4 GHz bands such as WiFi, ZigBee, Bluetooth, and others. Consequently, the radios have more reliable transmission with minimum interference from other common wireless devices and have much longer range with wireless data transmission up to 600 m in open air and over 150 m inside dense concrete buildings. Another popular wireless transmission protocol, Long Range (LoRa), was also initially considered, but the RFM69 FSK radios were ultimately chosen due in large part to their simplicity. To connect LoRa radios to the internet, a LoRaWAN gateway is required. Existing gateways can be used, but for applications where data is sensitive, deploying and maintaining a commercial gateway requires precise installation and configuration to be effective and is usually more expensive than their FSK counterparts. LoRa is also slower than FSK protocol due to reduced transmission bitrates. FSK radios typically have enough power and range for indoor applications, while LoRa radios are better suited for large-scale deployments like dense urban or forest settings where interval of transmission can be lower and longer range in the order of several miles is desired.

The range of the selected RFM69 radios was initially tested vertically in a dense high-rise commercial building in downtown Chicago, with successful data transmission between two floors (1400 m^2^ each), and laterally between two academic buildings with brick, glass, and steel construction, with successful data transmission from the core of one building to the core of another ~150 m away. The range can be further extended by decreasing the transmission baud rate of the radio and using antennas with higher gain. The RFM69 wireless radios have built-in hardware 128-bit AES encryption including CRC checks to ensure reliability and security of all data transmitted. Since the radio can act as a transmitter and a receiver, it can be used in both sensor nodes and control nodes. Coil antennas tuned to 915 MHz are used in sensor nodes located closer to the gateway. For sensor nodes that are located further away in the building, quarter-wave monopoles (a wire of approximately 80 mm in length) are used. For some nodes which are located in very dense buildings, an external whip antenna or a half-wave dipole antenna is used. In deployments with hundreds of nodes inside buildings, the gateway has an SAMD21 Cortex M0+ ARM processor (Microchip Technology Inc., Chandler, AZ, USA) with a larger ground plane, along with a rubber ducky or whip antenna with a high gain. There is also a transmission retry logic in each sensor and control node, which attempts to retry transmission of data if no acknowledgement is received from the gateway in a given interval of time. To avoid network congestion, this interval of time is randomized between 10 and 80 ms on each node independently to ensure that no two nodes attempt retrying transmission at the same time. A distributed backhaul setup also allows for multiple Raspberry Pis with gateways to be deployed in large buildings where the Pis are positioned to capture wireless data within a given region inside the building. The backend handles duplication of data and can process everything accordingly. The backend architecture is explained in further detail in Section 3.

Figure 3 shows the various components on a typical low-power sensor node (i.e., from Figure 2a). To achieve an idle current draw of 6.5 µA that allows for a long battery life, three things were done: (1) proper component selection, (2) software enhancements, and (3) using a switch to disable parts of the circuit.

Components on the board are specifically selected to minimize current draw. The voltage regulator has a very low quiescent current of 2 µA when active and the Schottky diode used for reverse polarity protection has extremely low reverse leakage current. In addition to the components, various power-saving features of the ATmega328P microprocessor (Microchip Technology Inc., Chandler, AZ, USA) are utilized by developing custom code. Since taking sensor measurements and transmitting data consumes the largest amount of current, it is more efficient to do it only intermittently when needed. The microprocessor’s power-saving sleep mode shuts off all internal circuitry except the important timer needed to wake itself up from sleep when desired. This timer is called a watch-dog timer and is an essential part of almost all embedded systems for various important tasks, especially where humans cannot easily access the hardware. The watch-dog timer’s sleep time is configurable on the ATmega328P and is set to a maximum of approximately 8 s. This means the processor will cease all activity and sleep for 8 s, then automatically wake up to continue where it left off. The firmware was specifically designed to have the processor wakeup and immediately sleep seven times in a row using its built-in timer before allowing it to take sensor measurements and transmit the data. This creates an interval of about 56 s per data transmission. The data transmission itself takes around 5–10 ms, while the sensor measurement is dependent on the type of sensor itself and is generally between 10–200 ms.

Most sensors also have some idle current draw when not actively measuring. To eliminate this, these sensors can be disabled during the 56 s sleep interval of the sensor node and enabled only when needed. This is done by means of an NPN transistor, which is a digital switch that allows the power line going to the sensor to be disabled. This can be used on those sensors that have quick start-up times and generally do not require any warm-up to measure accurately. The voltage drop from the transistor is typically not large enough to affect sensor performance. For some sensors that consume larger currents than the transistor can source and where voltage drop is not desirable, a MOSFET can be used instead.

The power source for all low-power sensors is chosen to be 3.7 V lithium ion polymer batteries. The discharge profile of a typical lithium ion battery is around 4.2 V when fully charged, dropping down to 3.7 V when being used. This simplifies the circuit by utilizing only one efficient step-down voltage regulator to provide a clean and regulated current at 3.3 V to all components. Using AA or AAA batteries was also considered, but combining at least 3 batteries to produce enough voltage for the voltage regulator would drastically increase the size and the weight of the sensor node, while using only one battery and a step-up converter to increase the voltage to a desired level would increase the idle current draw. Lithium ion batteries also have predictive behaviors whereby they are almost fully depleted by the time they drop below 3 V. This also allows for disabling an internal feature of the microprocessor called brown-out detection, which constantly monitors the operative voltage level during operation and if the voltage is dropped lower than 2.7 V, then it resets the chip. Since using lithium ion batteries ensures that voltage will never decrease below 3 V, the brown-out detection feature can be disabled, thereby saving additional power.

The PCB boards are designed in Autodesk EAGLE 9.1 and fabricated by OSHPark [33], a high quality board manufacturer for prototyping and light production based in USA. The board assembly is completed in house by our team. The firmware for each board is developed by our team in Arduino IDE 1.8.5. The library used for the radio is developed by LowPowerLab [34] and Adafruit’s open-source libraries were used for some sensors. Where no community-developed library was available, custom code was written by our team. Each board is programmed using the Pocket AVR Programmer [35] via the In-System Programming (ISP) header. This header is designed to take the Tag-Connect TC2030 cable [36] which allows for space-savings on a small sensor node. A custom jig built by our team is used to connect the cable to the programmer. The hardware and firmware files for the boards are open-source and are available on GitHub [37].

The red dotted box shown in Figure 3 indicates the part of the board where sensors are placed. Most sensor nodes utilize a similar board design, with components in the red box being replaced by other types of sensors to create nodes that measure various other things. Depending on the cost, measurement requirements, type of sensors, and final placement of the sensor node itself, one or more sensors can be combined into a single board, as shown in Figure 2e. Table 1 lists the components on the wireless sensor boards shown in Figure 2, along with their typical unit costs. The component costs vary depending on market supply and demand, distributer location and political factors. When sourcing components in quantity for mid-scale manufacturing, the component prices sharply decrease. Table 2 lists total cost for each of the wireless sensor nodes shown in Figure 2a–f, along with the backhaul.

For comparison, Table 3 lists total kit costs for deploying *Elemental* wireless sensors compared to some existing commercially available solutions, using a typical package of 3–5 environmental sensors and a single gateway. Note that total kit costs for both the commercial solutions and an *Elemental* package will vary based on the quantity of sensors or accessories included. The total kit costs for deploying a simple 5 node wireless temperature, humidity, and light package with a gateway is $120 USD for the *Elemental* platform, while similar commercial packages range from approximately $500 to $1900 USD. In addition to costs, other differences to consider include major differences in wireless communication protocols, power requirements, and availability of both local and cloud data storage/access. 

### 2.3. Gateway and Backhaul

Another fundamental part of the *Elemental* platform is a Raspberry Pi 3B, which is used as a central backhaul, shown in Figure 4. The Raspberry Pi is a miniature single-board computer with peripherals for internet connectivity running open-source software. The goal of this backhaul is to allow data monitoring and control of a multitude of devices over differing protocols in a single place. It captures all incoming wireless sensor data using gateways in the form of USB sticks that can be easily plugged in. A wireless gateway as shown in Figure 2f is configured to send and receive data via the RFM69HW radio module to allow communications with our wireless sensors and control nodes. Other existing gateways that are compatible with the backhaul are described in Section 2.4 and its firmware is described in Section 3.1.

### 2.4. Compatibility with Other Devices

In addition to connecting to sensor nodes, the *Elemental* platform can also connect to Smart Meter Connected Devices (SMCDs), which are verified by most service and utility providers to allow real-time building energy usage tracking from common smart meters. It is also compatible with consumer level and professional grade wireless weather stations. Gateways such as Meteostick from Smartbedded [38] can communicate with Davis weather stations and common weather sensor arrays from Ambient Weather (Chandler, AZ, USA), and is supported by the platform. This allows hyper-local weather monitoring immediately around the building to give granular and relevant data that can be used for analysis, instead of relying solely on weather stations located at airports which may not always take into account localized effects such as wind gusts, rain, or traffic. 

To allow full-home automation with existing IoT devices in the residential space, the *Elemental* platform is also compatible with existing wireless thermostats such as Radio Thermostat CT50 (Modesto, CA, USA); smart bulbs such as LIFX (Cremorne, Victoria, Australia); plug load monitors such as the Wemo Insight (Belkin, Playa Vista, CA, USA); and media players such as Google Chromecast (Menlo Park, CA, USA), with support for smart locks and other home-based IoT solutions currently being developed.

## 3. Software

The *Elemental* platform is an open-source Apache 2.0 licensed data monitoring, control, and automation system built with security and privacy in mind. It is designed from the ground up to run on the local network inside a building, without the dependence on internet connectivity. Connection to the cloud is optional for data backup, remote monitoring, or in-depth data analysis. This means that deployments of the platform are not locked into any one cloud service provider. However, for those systems that are already established on cloud service providers, the *Elemental* platform does allow remote deployment of the backend on AWS. The software part of *Elemental* consists of two parts: (1) a custom firmware for the Raspberry Pi developed in Elixir, and (2) a custom application for the backend called *Brood* that manages devices, provides a data store and enables real-time data visualization. All data transmitted between the backhaul and the backend are encrypted with high security protocols to ensure privacy.

### 3.1. Backhaul Firmware

The firmware on the Raspberry Pi is a custom-built embedded software based on the Nerves platform [39], written in Elixir language. Nerves allows development of fault-tolerant, self-repairing processes to ensure high data-logging reliability, which enables deploying lean, robust, and maintainable code on small embedded hardware. Nerves is used in various industrial products and allows for a scalable building data monitoring and control platform.

The *Elemental* platform offers a zero-configuration installation and will attempt to auto-discover many supported commercial IoT devices, along with our custom sensor and control nodes. Each device in the building discovered by *Elemental* is identified by a device ‘state’ by its device manager. Each state consists of device class, device IDs, a high-level name, device behaviour, and a few other parameters that help in identification and classification of the device. Device classification takes into account most building system devices, air quality sensors, and IoT devices, with room to customize and add more if needed. All device managing is done automatically and is fully self-repairing in the event of crashes or failures.

The *Elemental* platform consists of a distribution manager to allow multiple backhauls in a large commercial building to act as a mesh network. It also consists of a network manager that allows configuration of the local WiFi network inside the building on the Raspberry Pi backhaul. Figure 5 illustrates the overall architecture of the firmware on the Raspberry Pi. The firmware also has a local graphical user interface (GUI) that allows easy network configuration, historical data visualization, device control and system monitoring. The local GUI is explained further in Section 3.3.

Apart from allowing building data monitoring and control in both commercial and research applications, *Elemental* is also designed to be a full home/building monitoring and automation platform out of the box. The system works with a variety of consumer products for lighting, HVAC, media playback, energy monitoring, and weather monitoring as described in Section 2.4. To allow immediate installation of the platform, a pre-built firmware is provided on GitHub [37].

### 3.2. Elemental Backend

We developed a custom software package called *Brood* that provides all necessary infrastructure to allow hosting an independent local or cloud-based backend for deployment in buildings where security is critical. *Brood* uses Docker for infrastructure management. Docker is an enterprise-level container platform used widely by most major corporations for their applications [40]. This allows for secure, quick and easy replications of *Elemental* backend where needed. *Brood* consists of four parts:VerneMQ—MQTT (Message Queuing Telemetry Transport) message brokerInfluxDB—Timeseries databaseGrafana—Timeseries visualizationA middleware application to connect all components

Communication between the backend and the Raspberry Pi backhaul is done via encrypted MQTT protocol. MQTT is an open-source machine-to-machine IoT connectivity protocol, developed by OASIS (Burlington, MA, USA). It is an extremely lightweight publish/subscribe messaging transport protocol that is useful for connections with low-power devices in remote locations where a small code footprint is required and/or where network bandwidth is at a premium [41]. MQTT protocol has been successfully used in wireless heart-rate monitors [42], remote healthcare sensors for use in biomedical applications [43], various IoT projects [44] and many existing commercial systems. *Brood* uses VerneMQ, which is a high-performance, distributed MQTT message broker [45] to handle incoming and outgoing MQTT messages. *Brood* forces an encrypted MQTT connection between devices which requires the use of self-signed certificates to establish ‘trust’ between them.

InfluxDB is used for data storage and fast querying and processing. InfluxDB is an open-source time series non-relational database. It is written in Go language and has no external dependencies and is best suited for high-performance applications and scalability, especially for real-time data logging from a large number of wireless sensor nodes [46]. Studies have shown InfluxDB to be the best performing database for all applications regarding storage and querying of time-series data [47]. InfluxDB has also been used for intelligent environmental monitoring [48] and to demonstrate monitoring systems for large-scale smart city infrastructures [49].

Datalogging is divided into two types: real-time and interval-based data. Real-time data are event-based, e.g., motion in a room or a door opening. Interval-based data is any data point that is captured at a specified time interval, e.g., temperature or relative humidity every minute. Each data type is set to log in a separate InfluxDB database. Each InfluxDB data structure has a retention policy that describes how long InfluxDB needs to keep the data and delete data older than the defined duration. Based on our extensive testing, we have found that the total number of real-time events that occur in a typical indoor setting is not large enough, and hence takes less storage space, than originally expected. Thus, there is an unlimited retention policy defined in order to maintain full resolution of real-time data and preserve all historic data. On the other hand, interval-based data is stored at 1 min intervals, with a retention policy of one year. This data is also averaged every 15 min and 1 h, and stored in separate databases with unlimited retention policies, which allow for extremely fast querying of historical data over one year old and minimizes storage space in cloud-based applications.

While using *Brood* is not required for a local, offline deployment, it is recommended for long-term data storage, visualization and analysis. This offloads the computational work from the Raspberry Pi onto a dedicated system and allows for a more reliable data storage solution. With the flexibility of *Brood*, it can either be installed on a typical desktop PC for local access, or on a dedicated AWS instance for remote access over the internet and for high scalability and redundancy.

### 3.3. Data Visualization

*Elemental* provides two sources of data visualization: (1) a local GUI running on the Raspberry Pi backhaul, and (2) a cloud-based GUI running in *Brood*. The local GUI consists of a front-end dashboard written in Elm language. Elm is an open-source functional language for web-based applications. It is designed for simplicity and speed and has an extremely small footprint, which allows deployments on embedded hardware such as Raspberry Pis. It generates JavaScript for applications with no runtime errors and faster performance than other frameworks such as React, Angular, and Ember [50]. The dashboard is designed to be responsive, which allows it to work on mobile devices, laptops, and screens of any size. Figure 6 shows screenshots of the dashboard as viewed on a mobile phone.

The dashboard consists of graphs and histograms for each device type. This allows real-time monitoring of data from every single device on the network. The dashboard also allows controls for various IoT devices such as smart bulbs and media players, but also for various building systems such as thermostats and custom control nodes. For commercial deployments, *Elemental* provides a cloud-based solution for data monitoring through *Brood*, as explained in Section 3.2. Data visualization is done using Grafana, a leading open-source software for real-time data monitoring and time series analytics [51]. Grafana has been successfully used in air quality monitoring [52,53] and in energy monitoring in buildings [54]. Grafana uses the InfluxDB database in *Brood* as the data source and also exposes HTTP APIs to allow secure and easy retrieval of data for analysis without affecting the main database. Grafana also contains an administration interface for configuration of devices, as well as a user and organization management interface. This allows customized data access for users, depending on the application.

## 4. Results and Discussion

Three distinct applications of the *Elemental* platform are demonstrated here, including: (1) deployment in a research laboratory for long-term data collection and analysis, (2) a full-home monitoring solution, and (3) fault and anomaly detection and diagnostics of individual building systems at the zone-level.

### 4.1. Long-Term Data Collection and Analysis

*Elemental* has been deployed in a ~90 m^2^ research lab and office space on the campus of Illinois Institute of Technology for over 16 months, capturing data from various sensors placed around an academic building and logging at 1 min intervals. The backend exposes an HTTP API that allows various types of datasets to be extracted and analysed using simple queries to the server. Using this API, around 12 months of temperature, relative humidity, CO_2_ concentration, and occupancy data were extracted from the office space. The frequency distributions of 15 min averages of temperature, relative humidity, and CO_2_ concentration data from this deployment are shown in Figure 7a–c, respectively, along with a battery voltage profile of one of the sensor nodes (i.e., from Figure 2a) used in the deployment in Figure 7d. 

The frequency distributions show a wide range of indoor environmental conditions measured in the research lab, which is housed in an older academic building (built in 1946) and commonly experiences large swings in temperatures and relative humidity. In fact, temperatures ranged from a minimum of 15 °C to a maximum of 29 °C, with the bulk of the data ranging between 22 °C and 26 °C. Similarly, relative humidity ranged from a minimum of <10% to a maximum of 65%, with a bimodal distribution with one group centered around 20–35% and another centered around 50%, which represents winter and summer seasons, respectively. The vast majority of CO_2_ concentrations ranged 400–500 ppm, albeit with transient peaks as high as 1300 ppm. The sensor node shown in Figure 7d was powered using a rechargeable 1200 mAh lithium ion polymer battery and had a battery life of almost 8 months. A larger capacity battery (e.g., 2000 mAh) would extend its lifetime to well over one year. Additionally, exposure to some relatively large temperature changes (i.e., Figure 7a) is expected to have negatively affected battery performance.

The same indoor temperature, relative humidity, and CO_2_ concentration data from Figure 7 are also summarized on a monthly basis and shown in Figure 8a–c, respectively, along with weekly averaged occupancy data (measured using a PIR sensor node from Figure 2b) in the same space over the same period of time in Figure 8d. The general trends of indoor temperature and relative humidity clearly correlate to the seasonal changes in outdoor weather in Chicago, with November-April being the cold and dry months and May-October being warmer and more humid months. Mean indoor CO_2_ levels also correlate with occupancy (to some extent) and weather patterns (to a greater extent), which is consistent with the fact that the rooftop air handling unit, which provides only cooling and ventilation (but not heating), generally only operates in warmer months. During colder months, the building is heated by a steam radiator system and relies on infiltration for ventilation air. Weekly occupancy levels show typical activity in the office space during the year, with peak activity during the Spring and Fall semesters, and less frequent periods of high activity during the summer semester as students leave for vacation. Combined, the data in Figure 7 and Figure 8 demonstrate the utility of the *Elemental* platform in reliably collecting large amounts of indoor environmental data that can help better understand the spaces in which people live and work.

Additionally, since InfluxDB is specifically designed for handling time-series data, querying this amount of data is relatively quick, even on a slower spinning hard drive. This allows for fast data analysis when needed. An example of such data analysis is shown in Figure 9a. From the raw time-series data summarized in Figure 7, around three months (Feb-May) of daily CO_2_ data was extracted and shown in Figure 9a, with the blue line indicating the average diurnal CO_2_ trend during this time period. Next, a Python script was written to automatically analyse the data and extract the natural decay of built-up CO_2_ (generated by occupants) at the end of every day (Figure 9b). Air change rates (ACRs), or the rate of turnover or dilution of indoor air, from these decay data are then automatically calculated and plotted in Figure 9c. The trend in ACRs over this time period indicates their correlation with varying indoor-outdoor temperature differences as the weather changes from winter to spring, which is consistent with indoor/outdoor temperature differences and varying wind conditions as key driving forces of air infiltration in buildings [55]. This is just one example of how the *Elemental* platform can be used to collect large amounts of time-series data over long periods of time and how the data can be accessed for analysis and generation of useful information.

### 4.2. Full-Home Monitoring Solution

*Elemental* has also been deployed simultaneously in several apartments and homes around the city of Chicago, IL USA. To demonstrate the effectiveness of the platform, a full suite of hardware devices was provided to each resident. This included an air quality sensor (i.e., from Figure 2e), an energy monitor (RAVEn USB adapter, Rainforest Automation, Vancouver, BC, Canada, shown connected to the Raspberry Pi in Figure 4), wireless thermostat (Radio Thermostat CT50), a weather station (Ambient Weather WS-1000) and a Raspberry Pi backhaul. Figure 10a shows an overview of the data collected, including temperature, humidity, IEQ, barometric pressure, wind speed, solar radiation, rain fall, and energy usage, over a week in Grafana. Grafana also allows custom alerts to be set based on threshold values of measurements. The residents had individual access to data and providing them with insight about their homes led to increased awareness of poor air quality along with habits that lead to increased energy usage. Figure 10b shows eight months (August-March) of energy trends extracted from four apartments (each with areas between 100–180 m^2^). Some apartments showed typical energy use trends with higher energy consumption during summer and lower during winter due to reduced usage of forced air cooling as the weather changes, while others showed slightly higher energy usage in winter months, presumably due to increased usage of electric heating systems and artificial lighting during colder and shorter days.

### 4.3. Fault and Anomaly Detection and Diagnostics

Finally, the *Elemental* platform was also deployed in several additional faculty, staff, and student offices, classrooms, and conference rooms in the same university building at the Illinois Institute of Technology that was described in Section 4.1. Over 75 custom sensor PCBs, as shown previously, were deployed throughout these spaces to monitor indoor air quality and building system and thermal performance at the individual zone-level, including temperature sensors installed on the surface of steam radiators in each room to indicate utilization of the manually-controlled heating systems in each location. The data were also wirelessly transmitted to a local deployment of *Brood* in the previously described research laboratory space. Due to the availability of high-resolution data from the platform, the performance of each individual radiator could be analysed immediately with the intent of detection faults or anomalies in heating system performance. Figure 11 shows an example of 1 week of radiator surface temperature data that was monitored in several of these rooms. 

The arrows indicate those rooms where despite the control being shut off, the radiator temperature was significantly higher than room temperature, where all others dropped down to the ambient room temperature (denoted by the red arrows in Figure 11). Discovery of this anomaly, which was made feasible by the platform, led to a visual inspection of these outlier rooms that subsequently allowed for identification of faulty steam valves at those individual radiators that did not fully close and consequently leaked steam continuously, providing heat during all periods of the day and night even if occupants set their manual controls to the ‘off’ position. Identifying this fault allowed the maintenance team to immediately rectify the issue and save energy from being wasted further. Numerous other fault detection applications in various other building systems are possible in real-time with the use of the *Elemental* platform.

Additionally, this more widespread deployment also allowed for better understanding of potential data transmission, congestion, and packet loss issues. The only losses of data packets that occurred during this deployment were from those sensor nodes that were farthest from the gateway with numerous walls in between and that were most influenced by environmental conditions (e.g., the sensor nodes placed closest to the uninsulated brick exterior walls where extreme temperature changes occur during some very cold days in winter, which affected transmission performance). Fortunately, if delays or dropped data are experienced in a setup, more backhauls can easily be added.

## 5. Conclusions

Low cost sensor platforms play an important role in ubiquitous monitoring in various applications. This paper successfully demonstrates the use of low-cost open-source hardware and software to provide a comprehensive building monitoring solution. The *Elemental* platform is built from the ground up to address drawbacks of other similar platforms in terms of security, data storage, flexibility, deployment and low-power hardware. The platform also solves issues related to data-logging in scientific research projects which include limited storage, disaggregation of data, frequent check-ups to verify functionality, and others. It also aims to assist in easy data extraction of long-term data. The platform demonstrated here allows easy and virtually unlimited datalogging, monitoring, and analysis of real-time sensor data with minimal setup costs. Low-power sensor nodes placed in abundance in a building can also provide precise and immediate fault-detection, allowing for tuning equipment for more efficient operation and faster maintenance during the lifetime of the building. Smarter HVAC, VAV, and fan control is also possible with instant localized feedback from building zones, allowing for better thermal comfort and energy savings. For commercial installations we are also working on integrating BACnet and Modbus protocols that can communicate with existing HVAC systems. Due to the flexibility and cost-effectiveness of the platform, it can even enable large scale research studies involving indoor environmental quality, building energy performance, or city-wide traffic, air quality, and pedestrian tracking. The platform also enables long-term remote monitoring in those areas where physically accessing the sensor nodes repeatedly is not always practical. This opens the possibilities to allow air quality and environmental monitoring in underdeveloped areas of cities or in warehouses and storages. With the exponential development of Micro-Electromechanical Systems (MEMS) technologies in the last few decades, miniature pressure sensors, fluid accelerators, micro-scale energy harvesting devices and various other MEMS-based sensors can be directly be integrated with building mechanical, electrical and plumbing systems to provide real-time feedback where none was available before.

## Figures and Tables

**Figure 1 sensors-19-04017-f001:**
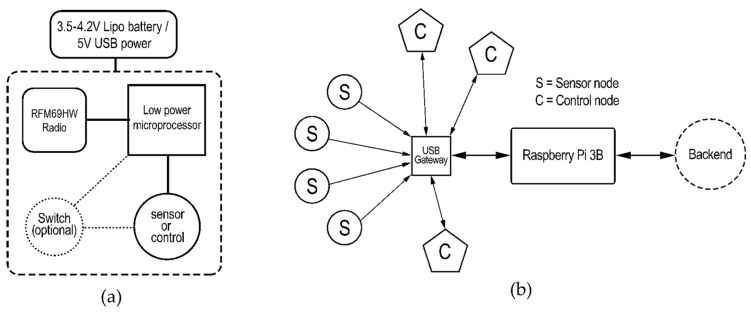
(**a**) Block diagram illustrating components in a basic wireless sensor/control node, (**b**) network architecture of a typical wireless sensor network.

**Figure 2 sensors-19-04017-f002:**
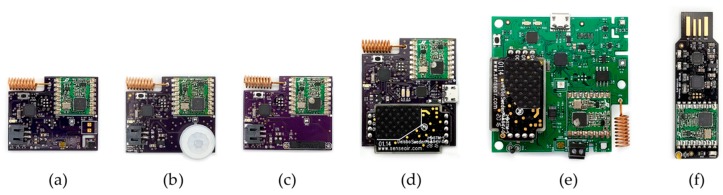
Low-power wireless sensor nodes measuring (**a**) temperature, relative humidity, and light intensity, (**b**) motion/room occupancy, (**c**) door/window opening, and (**d**) CO_2_ concentration; (**e**) an IAQ monitor with extended capabilities; and (**f**) USB gateway receiver for all wireless sensor nodes.

**Figure 3 sensors-19-04017-f003:**
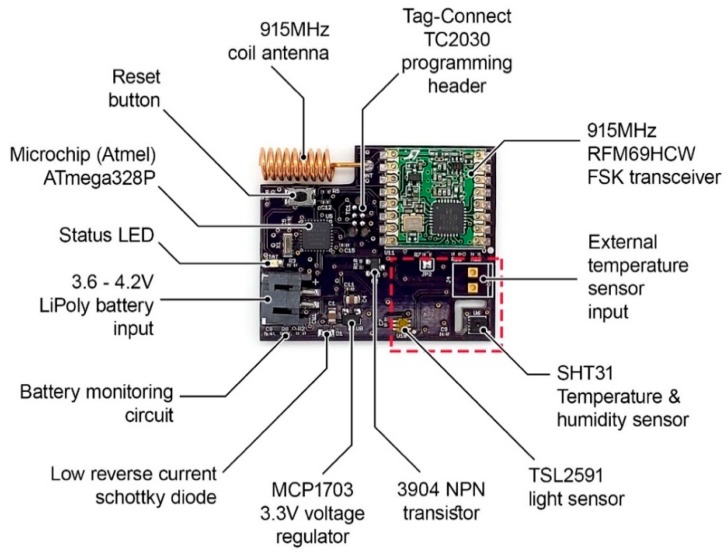
Various components of a typical wireless sensor node (in this case, the node is the same as in Figure 2a and includes temperature, relative humidity, and light sensors).

**Figure 4 sensors-19-04017-f004:**
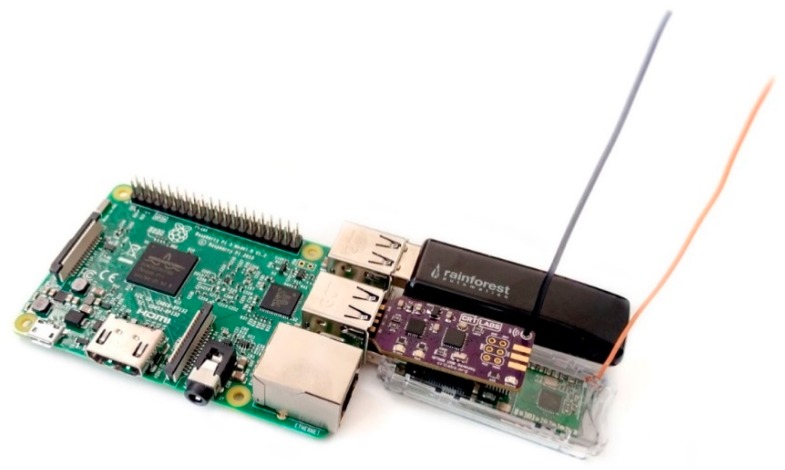
Raspberry Pi backhaul with gateways to connect with various wireless devices.

**Figure 5 sensors-19-04017-f005:**
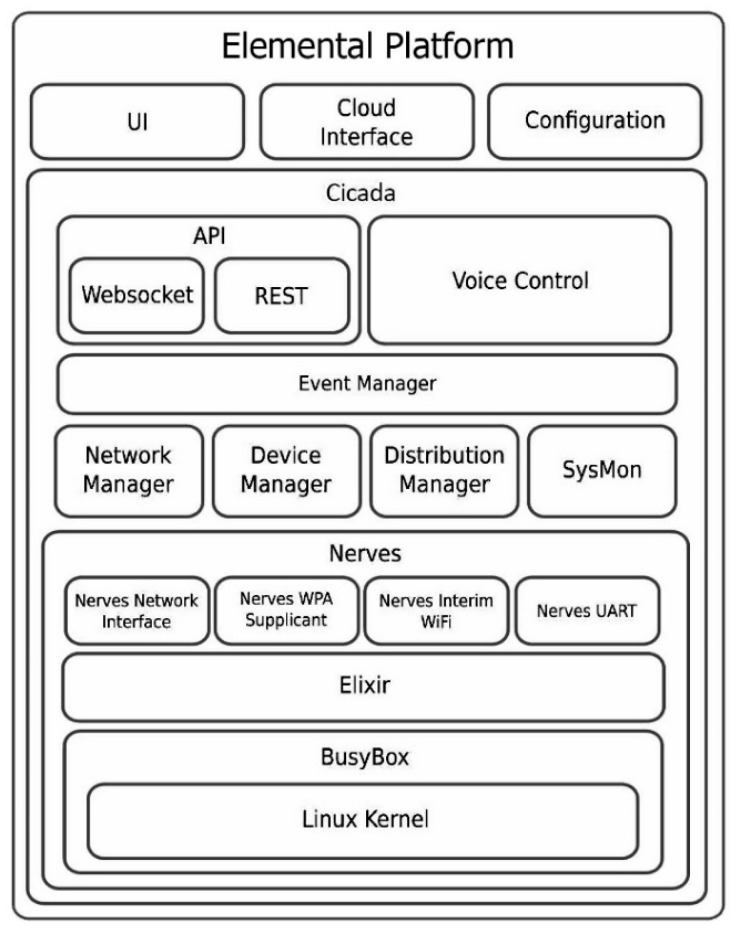
Elemental platform firmware architecture.

**Figure 6 sensors-19-04017-f006:**
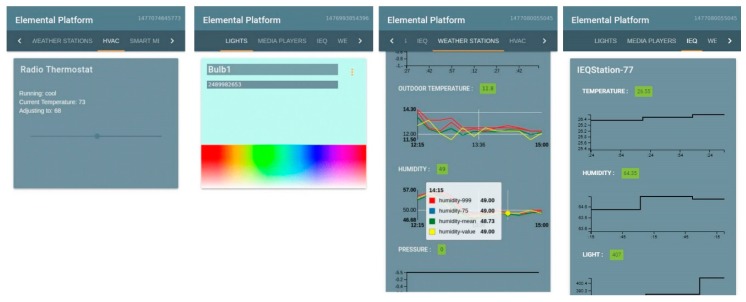
Various screens of the *Elemental* local dashboard running on a Raspberry Pi.

**Figure 7 sensors-19-04017-f007:**
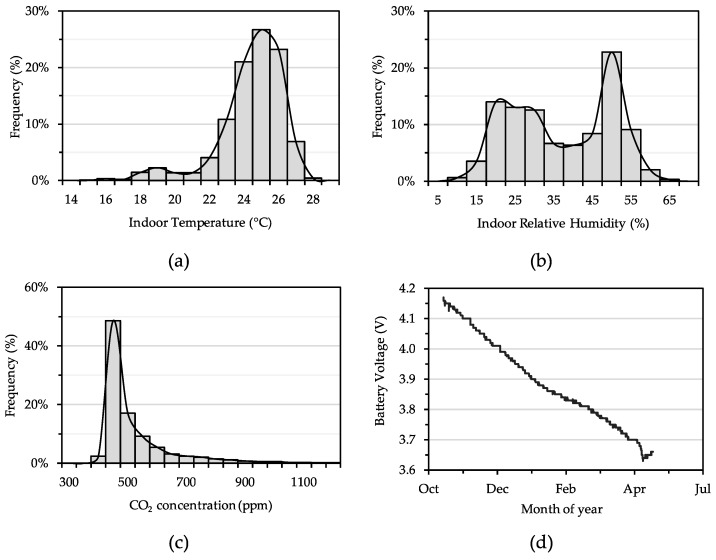
Histograms showing frequency distributions of 15-min averages of indoor (**a**) temperature, (**b**) relative humidity, and (**c**) CO_2_ concentration, and (**d**) initial battery life (voltage over time) for a sensor node deployed in a research lab and office space for ~12 months.

**Figure 8 sensors-19-04017-f008:**
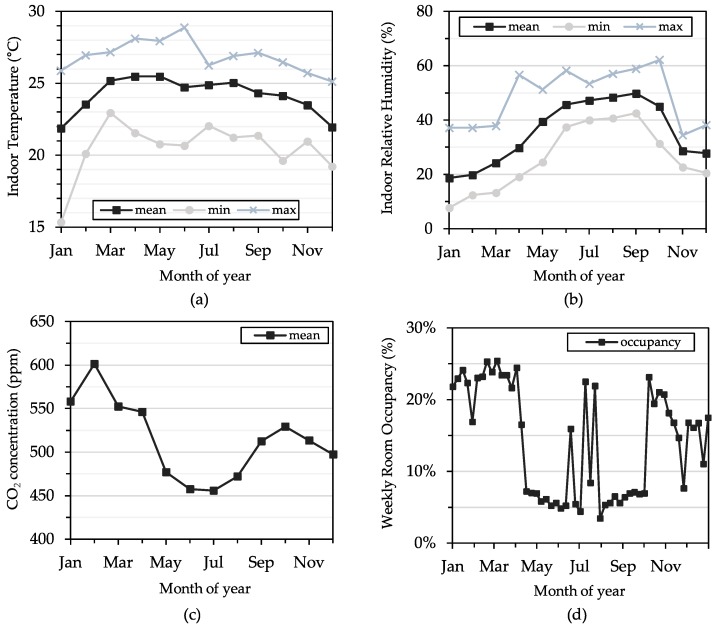
Annual trends showing monthly (**a**) temperature, (**b**) relative humidity, (**c**) CO_2_ concentration, and (**d**) weekly occupancy percentage in a research lab and student office space in an academic building.

**Figure 9 sensors-19-04017-f009:**
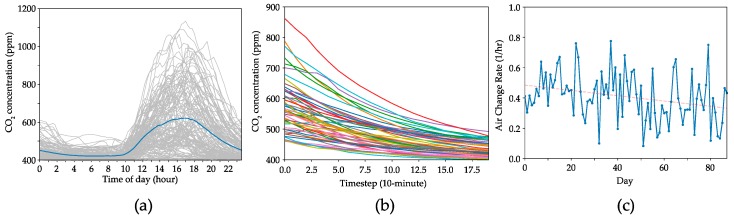
Data from the *Elemental* platform deployed in an academic lab and office space showing (**a**) daily CO_2_ concentrations, (**b**) extracted natural CO_2_ decay curves, and (**c**) air change rates calculated from the decay curves each day, plotted versus time.

**Figure 10 sensors-19-04017-f010:**
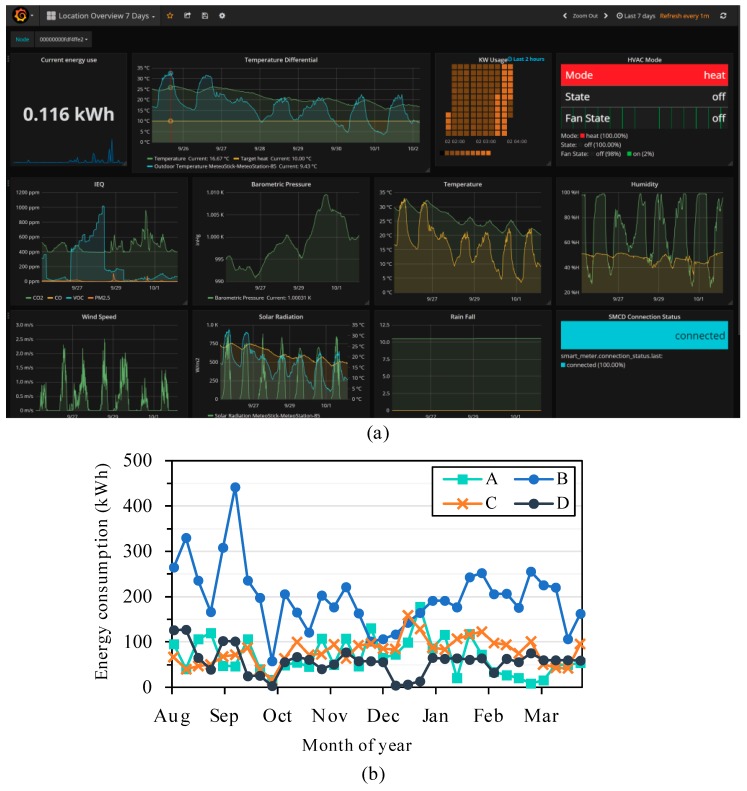
(**a**) Grafana—data monitoring platform in *Brood* used for monitoring air quality, energy and building systems in a residential setting and (**b**) 8 months of weekly energy consumption data in four apartments.

**Figure 11 sensors-19-04017-f011:**
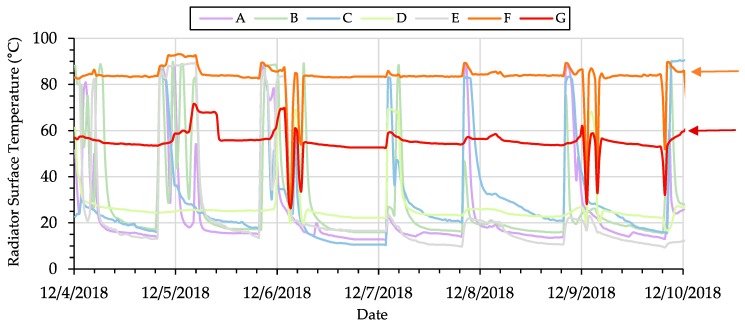
One week of radiator surface temperature data revealed a fault in the system (red arrows). Lines labeled A through G indicate individual radiators, each in separate rooms in the building.

**Table 1 sensors-19-04017-t001:** List of components in the *Elemental* wireless sensor board family.

Components	Part Number	Unit Cost (USD)
Common parts		
Microprocessor	Microchip ATmega328P	$2
Voltage regulator	MCP1703-3.3 V	$0.50
Radio	RFM69HCW-915 FSK Transceiver	$4
Switch	MMBT3904 NPN transistor	$0.10
Battery	Lithium-Ion polymer (1200 mAh)	$5
Sensors		
Temperature & humidity	Sensirion SHT31	$6
Light	AMS TSL2591	$2
Motion	Parallax mini PIR sensor	$10
Door/Window opening	Soway NO Reed switch	$0.50
Carbon Dioxide	SenseAir S8	$85
TVOC	Sensirion SGP30	$12
PM_2.5_ and PM_10_	Plantower PMS7003	$15
Barometric Pressure	Bosch BMP388	$3
Surface temperature	US sensors PR103J2 precision thermistor	$6
Other components		
USB interface	FTDI FT231XQ USB 2.0 full speed IC	$2
Flash memory	Windbond 4mbit W25 × 40CLSNIG	$0.40
Coil antenna	915 MHz helical coil antenna	$0.10
Rubber ducky antenna	915 MHz 3dBi SMA antenna	$2.50

**Table 2 sensors-19-04017-t002:** Total unit cost of *Elemental* wireless sensor family and backhaul.

Board Type	Unit Cost (USD)
Elemental wireless sensor boards	
Temperature, relative humidity, light intensity node	$15
Occupancy/motion node	$20
Door/window node	$10
CO_2_ concentration node	$40
All-in-one IAQ node	$85
Wireless USB Gateway	$10
Backhaul (including supported USB gateways)	
Raspberry Pi 3B+	$35
Rainforest Automation RAVEn USB adapter	$40
Smartbedded Meteostick	$178

**Table 3 sensors-19-04017-t003:** Comparative unit and total cost of deployment of existing commercially available wireless solutions versus the *Elemental* wireless sensor family.

Name	Unit Cost (USD)		Total Kit Cost (USD)
	Sensor	Gateway	
**Onset HOBO ZW Series wireless monitoring kit (3 sensors, 1 gateway)**	$274	$219	$989
Monnit Wi-Fi temperature monitoring bundle (3 sensors)	$159	-	$487
TempGenius complete 5 sensor system	-	-	$1,899
Lascar EL-WiFi-TH-Plus (3 sensors)	$200	-	$600
*Elemental* wireless T/RH/Light 5 sensor kit	$15	$45	$120

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
