# Peer review of "Elemental: An Open-Source Wireless Hardware and Software Platform for Building Energy and Indoor Environmental Monitoring and Control"

_sensors, 2019, doi:10.3390/s19184017_

Round 1

Reviewer 1 Report

This is an interesting paper, presenting an open source approach for a complete solution to the problem of monitoring indoor environmental parameters inside buildings. The authors discuss in detail their hardware and software implementation, with very interesting aspects presented in the paper.

One issue that I have with the paper is its title: at first sight, when seeing the "building energy" part, I imagined that the open source aspect would extend to the provision of energy monitors as well. However, this is not the case, since the authors rely on other devices to get such data, while for the rest of the environmental parameters they present designs of their own. For this reason, I find it a bit misleading to include the term "building energy" part in the title. For the control part, the authors showcase a custom PCB with such capabilities.

I also would like to see a larger focus on calibration: although it is a big issue in such applications, it is not mentioned much in the text. However, the sensors mentioned in the text are generally compatible with what is currently used by the community and of acceptable quality.

Regarding the radios used, I think a good idea would be to add some comparison/argumentation with respect to other similar technologies like LoRa which is a valid choice for this type of applications as well. I would also like to see more details on the performance of the radio. E.g., how do these radios perform inside buildings but in different floors, or with different types of material (walls, etc.) between them? The authors state that Elemental has been deployed in several apartments etc. (line 451), but do not get any description to what those buildings like, distance between nodes, number of hops, and so on and so forth. This would enhance the paper in my opinion.

I would suggest to add the following 2 references, both good examples of open source hardware design related to the work presented in this paper, and which also tackle related issues a bit more in detail. The first work presented an open source approach that could be used in indoor and outdoor environments in smart cities and stresses the calibration and data quality issues, while the second one deals with energy monitoring aspects explicitly:

Guillem Camprodon, Óscar González, Víctor Barberán, Máximo Pérez, Viktor Smári, Miguel Ángel de Heras, Alejandro Bizzotto, Smart Citizen Kit and Station: An open environmental monitoring system for citizen participation and scientific experimentation, HardwareX, Volume 6, 2019. Lidia Pocero, Dimitrios Amaxilatis, Georgios Mylonas, Ioannis Chatzigiannakis, Open source IoT meter devices for smart and energy-efficient school buildings, HardwareX, Volume 1, Pages 54-67, 2017.

The paper overall is well-written, with a clear structure and very few mistakes (e.g., on line 291, "any many supported"). The figures are useful and easily legible. Some minor mistakes with the references as well, e.g., in line 310 perhaps the github repository reference is wrong.

Moreover, from what I understand the GitHub repository of the project is being enhanced this week - some additional details/examples would be welcome in the documentation.

Also, as a final comment, although I understand that this paper required a lot of work to come to completion, in some parts in the description of the software it is not always clear what the authors did provide over existing solutions. Maybe it would be a good idea to make their contributions a bit more clear.

Overall, I enjoyed reading this work and  I propose to accept the paper with minor revision, after the comments above have been addressed.

Reviewer 2 Report

This paper presents the Elemental platform-a low-cost open-source hardware and software to provide a comprehensive building monitoring solution. Many advantages are found from the work, however, it can be improved to enhance the quality and comparative study. For example, how the cost of proposed hardware (and/or including software)? How is it compared to the existing solution? The other concern is about the performance of network connecting a large number of nodes in term of energy, delay when deploying routing protocols?
